# Accurate localization microscopy by intrinsic aberration calibration

Craig R. Copeland [1], Craig D. McGray [2], B. Robert Ilic [1,3], Jon Geist[2] & Samuel M. Stavis [1]✉

A standard paradigm of localization microscopy involves extension from two to three dimensions by engineering information into emitter images, and approximation of errors resulting from the field dependence of optical aberrations. We invert this standard paradigm, introducing the concept of fully exploiting the latent information of intrinsic aberrations by comprehensive calibration of an ordinary microscope, enabling accurate localization of single emitters in three dimensions throughout an ultrawide and deep field. To complete the extraction of spatial information from microscale bodies ranging from imaging substrates to microsystem technologies, we introduce a synergistic concept of the rigid transformation of the positions of multiple emitters in three dimensions, improving precision, testing accuracy, and yielding measurements in six degrees of freedom. Our study illuminates the challenge of aberration effects in localization microscopy, redefines the challenge as an opportunity for accurate, precise, and complete localization, and elucidates the performance and reliability of a complex microelectromechanical system.

---

[1] Microsystems and Nanotechnology Division, National Institute of Standards and Technology, Gaithersburg, MD, USA. [2] Quantum Measurement Division, National Institute of Standards and Technology, Gaithersburg, MD, USA. [3] CNST NanoFab, National Institute of Standards and Technology, Gaithersburg, MD, USA. ✉email: samuel.stavis@nist.gov

Microscopic objects have structure and motion in three spatial dimensions and six degrees of freedom. Whereas classical implementations of optical microscopy resolve images in only two dimensions, recent advances enable localization of the positions of single emitters in all three dimensions[1]. Such measurements typically involve custom optics to encode aberrations that vary predictably as a function of position along the optical axis and decoding axial positions from the resulting lateral images. This engineering approach can improve some metrics of microscopes while degrading others, within theoretical limits[1,2], and has practical limitations. Models of microscopes are imperfect and nontrivial to develop[1,3], discouraging microscopists who focus on applications rather than instrumentation. Custom optics add complexity to the integration and alignment of microscope systems, degrade localization precision by reducing the transmission of signal photons to the imaging sensor, and degrade localization accuracy by increasing unpredictable errors from aberration effects[4–7]. For this latter reason, engineering approaches require at least estimation of localization errors, if not calibration to correct the errors. However, such analysis is uncommon in practice, resulting in a common discrepancy between precision and accuracy that can approach a factor of four orders of magnitude across a wide field[8].

Many applications can benefit from precise and accurate localization of single emitters in three dimensions[2]. We consider two applications that bracket a wide range of experimental complexity, extending the scope of the measurement to tracking multiple emitters as indicators of the six degrees of freedom of microscale bodies. These two aspects of this complete measurement are synergistic, as multiple emitters improve localization precision and orientation precision by rigid transformations that combine information through the central limit theorem, while the rigidity and planarity of a microscale body enable tests of tracking accuracy[9–12]. Toward the simple end of the application range, the deposition of fluorescent particles on an imaging substrate – a microscale body that is ubiquitous in localization microscopy – allows calibration of aberration effects[5,6,13,14] and correction of instrument drift[15–17]. The interest in performing localization microscopy within macroscopic volumes[18–20] introduces challenges of leveling samples and imaging them through focus. Toward the complex end of the range of applications, the coupling of microscale bodies within a microsystem controls the output of force and motion to perform work. This essential function of machines has diverse applications to optical traps[21], colloidal motors[22], tunable photonics[23,24], reconfigurable metadevices[25], materials characterization[26,27], and even safety switches of extreme consequence[28]. The latter application is exemplary of microsystem technologies that integrate multiple parts, are nominally planar, and benefit from tracking in two dimensions and three degrees of freedom to elucidate their motion[9,11,12,29,30]. However, measurements in three dimensions[31] and six degrees of freedom are much more informative[32].

In the present study, we demonstrate that comprehensive calibration of the effects of intrinsic aberrations of an ordinary microscope enables precise and accurate tracking of single emitters in three dimensions throughout an ultrawide[6] and deep focal volume. This concept makes use of the latent information of intrinsic aberrations, avoids custom optics, maximizes signal photons, and preserves the intrinsic lateral extent of the point spread function[33]. We exploit intrinsic astigmatism and defocus among other aberrations to localize multiple emitters in three dimensions on an imaging substrate and on a complex microsystem[34,35], extending the concept to measurements of motion in six degrees of freedom (Fig. 1). The development and application of our method are synergistic, as the microsystem

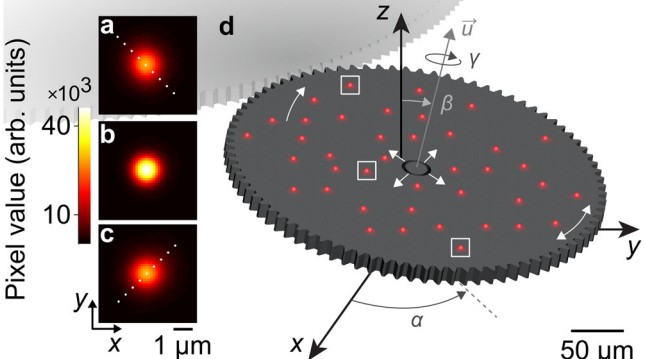

**Fig. 1 Intrinsic aberrations enable accurate localization microscopy in six degrees of freedom. a–c** Fluorescence micrographs showing images of a particle at z positions of (**a**) 2 μm above, (**b**) near, and (**c**) 2 μm below best focus. The particle diameter is 1 μm and the resolution limit is 0.7 μm. Two aberration effects are apparent – symmetry variation from astigmatism and intensity variation from defocus. Dots indicate asymmetry in (**a**, **c**). Vertical positions correspond to white boxes in (**d**). **d** Schematic showing (red) fluorescent particles on part of a complex microsystem. We localize single particles in three dimensions and fit a rigid transformation to measure motion with six degrees of freedom – translations $\Delta_x$, $\Delta_y$, and $\Delta_z$, intrinsic rotation $\gamma$ about the axis of rotation $\vec{u}$, nutation $\beta$, and precession $\alpha$. White arrows indicate play due to clearances in the microsystem. (**d**) Lateral dimensions are nearly to scale. Vertical dimensions are not to scale.

functions both as a rotary microstage to rigorously test the accuracy of position data in three dimensions, and as a device under test with critical kinematics in six degrees of freedom to elucidate. Even for the slight aberrations that remain in a modern microscope after optical engineering to correct them, our method achieves axial precision of 25 nm and axial range of 10 μm, and lateral precision of 1 nm and lateral range of 250 μm, at a frequency of nearly 100 Hz. These performance metrics, and the ability to measure motion in six degrees of freedom with an ordinary optical microscope and simple localization analysis, distinguish our method from more specialized combinations of microscopy and interferometry (Supplementary Table 1)[31,36–39]. Just as importantly, our method achieves accuracy that is commensurate with precision, by calibrating magnification[8,40] and the field dependence of aberration effects that cause localization errors[4,8]. Most importantly, our study illuminates a fundamental problem – intrinsic aberrations deform imaging fields of surprisingly small extent, causing errors which can require widefield calibration and axial localization to achieve lateral accuracy that is truly better than the imaging resolution[4]. Our method provides not only a practical solution to this problem but also the opportunity to exploit intrinsic aberrations for localization microscopy in all three dimensions and six degrees of freedom.

## Results and discussion

**Overview of method.** Whereas localization precision requires signal photons, localization accuracy requires microscope calibration. Random arrays of subresolution particles enable characterization of the point spread function and registration of localization data from different wavelengths[5,6,13,14,41,42]. Regular arrays of subresolution apertures allow calibration of magnification and distortion[8], and other aberration effects on localization[4,8]. Random arrays of molecular nanostructures provide reference positions to determine local magnification[43,44]. However, no study has completely calibrated a localization microscope. We approach this closer than before by integrating information from two types of emitter arrays. We image fluorescent microparticles and subresolution apertures through focus

(Fig. 1, Supplementary Fig. 1) and fit bivariate Gaussian models to the images,

$$G_{biv}(x_p, y_p) = A \cdot \exp\left(-\left(\frac{1}{2(1-\rho^2)}\left[\frac{(x_p-x')^2}{w_x^2} - 2\rho\frac{(x_p-x')(y_p-y')}{w_x w_y} + \frac{(y_p-y')^2}{w_y^2}\right]\right)\right) + B$$

(1)

where $x_p$ is the position of a pixel in the $x$ direction, $y_p$ is the position of a pixel in the $y$ direction, $A$ is the amplitude, $x'$ is the apparent position of an emitter in the $x$ direction, $y'$ is the apparent position of an emitter in the $y$ direction, $w_x$ is the standard deviation in the $x$ direction, $w_y$ is the standard deviation in the $y$ direction, $\rho$ is the correlation coefficient between the $x$ and $y$ directions, and $B$ is a constant background. The image shapes, dependences on field position, and purposes of the two types of emitters all differ. Particle arrays enable calibration of image shape for axial tracking of experimental particles, and aperture arrays enable calibration of magnification and distortion for conversion of units from pixels to nanometers. The apparent lateral positions and image shapes of both types of emitters vary with axial position. The emitters sample the field at discrete locations. At each location, continuous functions in one dimension model the axial dependences of apparent lateral position and image shape. For widefield calibration, continuous functions in two dimensions model the lateral dependence of apparent lateral position and image shape (Supplementary Table 2, Supplementary Fig. 2).

**Axial dependence of aberration effects**. We emphasize a critical result that is fundamentally problematic for super-resolution. Intrinsic aberrations affect apparent lateral positions, causing systematic errors that depend on axial position (Fig. 2a)[4,8,45]. These errors approach the imaging resolution, rendering much smaller values of localization precision potentially meaningless or even misleading. To achieve lateral localization accuracy that is truly superior to the imaging resolution, both axial localization and complete calibration of the field dependences are potentially necessary. Fortunately, intrinsic aberrations also encode axial information into emitter images, providing a latent capability for axial localization.

We develop this latent capability into a general and practical solution. For each calibration particle, empirical polynomials of high order model changes in $x'$ and $y'$,

$$\Delta_x'(z) = x'(z) - x'(z_f)$$

(2)

and

$$\Delta_y'(z) = y'(z) - y'(z_f)$$

(3)

where $z_f$ is the $z$ position of best focus (Fig. 2b, c). Values of $z_f$ occur at peaks of $A(z)$ (Fig. 2d)[8], defining the focal surface of $z_f \equiv 0$. Local calibration functions enable correction of $x'$ and $y'$ to their values at $z_f$ and require measurement of $z$. The uncertainty of local calibration (Fig. 2b, c, e, f, bottom plots) is the first of two main components of uncertainty in our study. We report uncertainties as 68 % coverage intervals, corresponding to ± one standard deviation or ± one standard error, depending on the context and accounting for a large number of replicate measurements, or we note otherwise. Lateral accuracy also depends on field curvature, lateral drift of the microscope system, and uncertainty of the independent variable for calibration of apparent lateral motion (Supplementary Note 1).

Away from $z_f$, intrinsic astigmatism causes asymmetry, and defocus decreases amplitude and increases width, of emitter images[46]. In a bivariate Gaussian fit, these effects manifest as variation of $\rho$, $A$, $w_x$, and $w_y$ (Figs. 1a–c and 2d, e, Supplementary Fig. 1). Common implementations of extrinsic astigmatism involve alignment of the axes of a cylindrical lens to the axes of

an imaging sensor, encoding axial information into variation of $w_x$ and $w_y$[47–49]. In contrast, we forgo any optical engineering or even careful alignment of our microscope system in a practical approach to extracting more information from the default data and analysis.

The bivariate Gaussian model approximates the image loci as ellipses with axes that, for our microscope, are at an angle of $\pi/4$ radians with respect to the $x$ and $y$ axes of the imaging sensor, and with eccentricity that varies with $z$ position. Over an axial range of a few micrometers, $\rho$ has unique values with a nearly linear dependence on $z$ position, due to intrinsic astigmatism. $A$, $w_x$, and $w_y$ also depend on $z$ position, due to defocus (Fig. 2d, Supplementary Fig. 1), but for our microscope the dependences are nearly symmetric above and below $z_f$. To break this symmetry and deepen the axial range, we define parameters for astigmatic defocus,

$$\rho_w = \rho \cdot \frac{|w_x| + |w_y|}{2}$$

(4)

(Supplementary Fig. 4) and

$$\rho_A = \frac{\rho}{A_n}$$

(5)

(Fig. 2f), where

$$A_n = A/A_{\rho=\rho_0}$$

(6)

is the amplitude after normalization to its value in the image for which $\rho = \rho_0$. We set $\rho_0$ to the minimum value of $|\rho|$. This ensures that axial dependence is independent of any differences of emission intensity between calibration and experiment. A flatfield correction[8] accounts for lateral nonuniformity of both illumination and detection, and calibration of axial localization accounts for axial nonuniformity of illumination. The uncertainty of local calibration, $\sigma_z$, is due mostly to the shot noise of a large number of signal photons and is the first of two components of uncertainty of axial localization. To quantify localization performance, we must consider the field dependences.

**Field dependence of aberration effects**. Calibration particles provide sets of inverse functions $\{z(\rho)\}_{cal}$, $\{z(\rho_w)\}_{cal}$, and $\{z(\rho_A)\}_{cal}$ (Supplementary Table 2, Supplementary Fig. 2), enabling axial localization across a wide lateral field. The inverse functions vary significantly, unpredictably, and systematically with lateral position, requiring calibration of field dependences and evaluation of $\sigma_z$ (Fig. 3, Supplementary Fig. 5). In contrast, uncertainties from $\Delta_x'(z)$ and $\Delta_y'(z)$ are nearly independent of lateral position (Supplementary Fig. 6). Each parameter has a unique utility, with $\rho$ resulting in good precision within approximately 1 μm of best focus, $\rho_w$ extending the axial range, and both parameters being independent of emission intensity. The latter property enables robust calibration in the presence of illumination nonuniformity and absence of flatfield correction, or in case of photobleaching. Any effects of photobleaching are insignificant in our experiment, as we show subsequently. For a constant emission intensity, $\rho_A$ optimizes precision, range, and uniformity, achieving minimum values of $\sigma_z \approx 25$ nm, local values of $\sigma_z \approx 30$ nm across much of the field, mean values of $\sigma_z \approx 40$ nm across the full lateral range and through most of an axial range of 6 μm, and 68 % intercentile ranges of less than 20 nm (Fig. 3). In light of this surprisingly high performance, we select $\rho_A$ for application.

Having a comprehensive evaluation of local precision, we develop our widefield method of first using the field dependence of $\rho_A$ for axial localization, and then using axial position to correct apparent lateral position. For a value of $\rho_A$ from lateral localization of an experimental particle, $\{z(\rho_A)\}_{cal}$ returns values of $z$ for the calibration particles. These $z$ values determine the apparent lateral

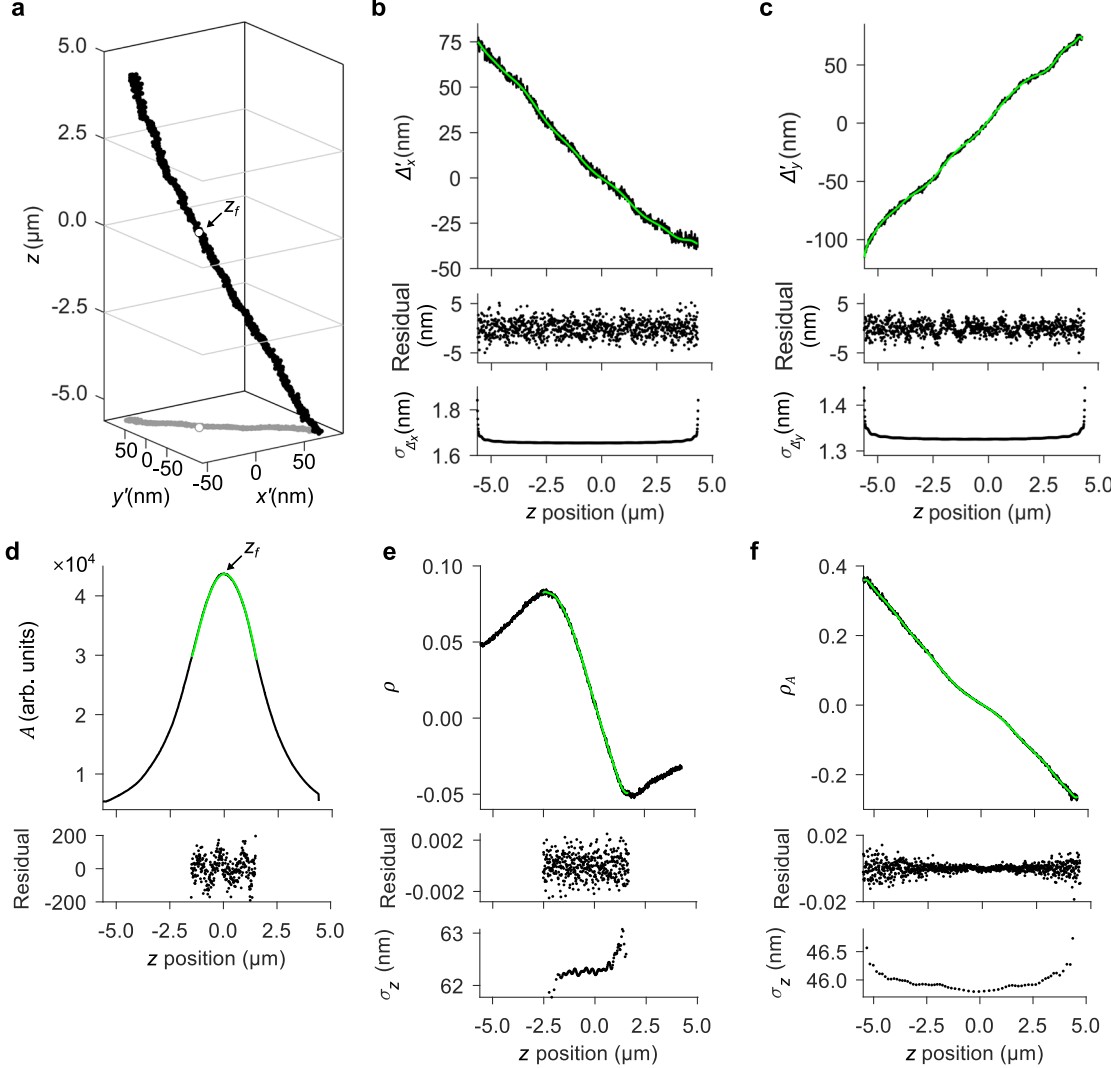

**Fig. 2 Effects of intrinsic aberrations on apparent lateral position and particle image shape.** These data are from a representative calibration particle at a representative location in the imaging field. **a** Scatter plot showing apparent lateral position as a function of actual axial position. White data markers indicate the actual lateral position, which we define at the axial position of best focus $z_f$. Uncertainties are smaller than the data markers. **b**–**f** Plots showing the dependence on axial position of the parameters (**b**) $\Delta'_x(z) = x'(z) - x'(z_f)$, (**c**) $\Delta'_y(z) = y'(z) - y'(z_f)$, (**d**) $A$, (**e**) $\rho$, and (**f**) $\rho_A = \frac{\rho}{A_n}$, where $A_n = A/A_{\rho=\rho_0}$ is the amplitude after normalization to its value in the image for which $\rho = \rho_0$, with $\rho_0$ set to the minimum value of $|\rho|$. Fits of bivariate Gaussian models to emitter images determine the (black data markers) parameter values, and (green lines) polynomials model the $z$ dependence for (**b**–**c**) lateral correction, (**d**) determination of the axial position of best focus $z_f$, and (**e**–**f**) axial localization. Residual values indicate an uncertainty for each parameter. Values in the bottom panels are uncertainties of (**b**-**c**) apparent lateral position $\sigma_{\Delta'_x}$ and $\sigma_{\Delta'_y}$ from the polynomial models, and (**e**-**f**) $z$ position $\sigma_z$ from inversion of the polynomial models.

positions of the calibration particles by $\{x'(z)\}_{cal}$ and $\{y'(z)\}_{cal}$, discretely sampling $(x', y', z)$ throughout the focal volume for the experimental value of $\rho_A$. Fitting these data with a continuous function $z(x', y'; \rho_A)$ yields a widefield calibration for that value of $\rho_A$, providing the axial position $z$ of an experimental particle at any apparent lateral position $(x', y')$ (Fig. 4a–c). Following axial localization, the $z$ position of an experimental particle enables correction of its apparent lateral position, beginning with local calibration functions for apparent translation, $\{\Delta'_x(z)\}_{cal} = \{x'(z) - x'(z_f)\}_{cal}$ and $\{\Delta'_y(z)\}_{cal} = \{y'(z) - y'(z_f)\}_{cal}$ (Supplementary Table 2, Supplementary Fig. 2). Like image shape, apparent translation varies significantly, unpredictably, and systematically with lateral position (Fig. 4d–f, Supplementary Fig. 7). Widefield calibration functions $\Delta'_x(x', y'; z)$ and $\Delta'_y(x', y'; z)$ determine the apparent translation of the experimental particle for the value of its axial position $z$ and at its apparent lateral position $(x', y')$.

The final position of the experimental particle is $(x = x' - \Delta'_x, y = y' - \Delta'_y, z)$. We calibrate an axial range of 6 μm across the full lateral field. Correction for tilt, or nutation, of the surface normal of the calibration substrate relative to the optical axis (Supplementary Fig. 8, Methods) leaves an axial range of 4 μm, which is sufficient for our application.

The selection and optimization of a widefield calibration function for $\rho_A$ are nonobvious. The purpose of this function is to accurately model astigmatic defocus throughout the field, on the basis of a discrete and finite sampling. However, axial dependences of $\rho_A$ can vary significantly across lateral regions of only a few micrometers (Fig. 4). This result emphasizes another fundamental problem for super-resolution, calling into question common expectations and approximations of uniform fields far beyond that length scale, and potentially requiring a widefield calibration for fields that are surprisingly small. Arrays of

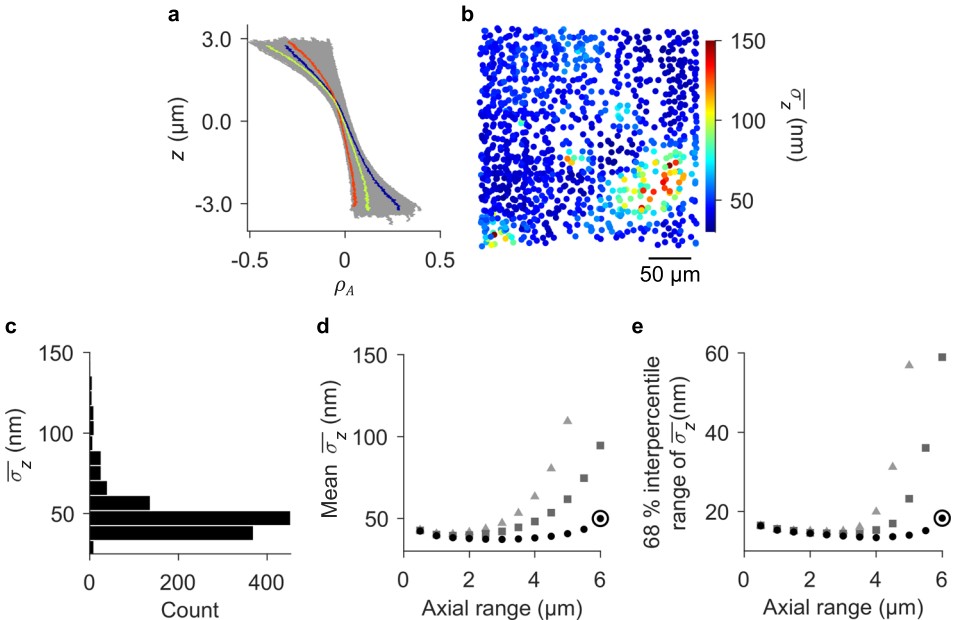

**Fig. 3 Field dependence of local values of uncertainty components from axial localization. a** Line plots showing the relationship $z(\rho_A)$ for many calibration particles. Three representative lines have colors corresponding to the map in (**b**) for local values of uncertainty $\bar{\sigma}_z$ from the polynomial models $\{z(\rho_A)\}_{cal}$. The overbar denotes the mean uncertainty over the axial range of (**a**). **b** Scatter plot showing the lateral positions of the calibration particles and corresponding values of $\bar{\sigma}_z$ for an axial range of 6 μm. **c** Histogram showing an asymmetric distribution of $\bar{\sigma}_z$ for an axial range of 6 μm. **d** Plot showing variation of the mean value of $\bar{\sigma}_z$ for all particles as a function of axial range for (triangles) $\{z(\rho)\}_{cal}$, (squares) $\{z(\rho_w)\}_{cal}$, and (circles) $\{z(\rho_A)\}_{cal}$. **e** Plot showing variation of the 68 % interpercentile range of $\bar{\sigma}_z$ for all particles as a function of axial range for (triangles) $\{z(\rho)\}_{cal}$, (squares) $\{z(\rho_w)\}_{cal}$, and (circles) $\{z(\rho_A)\}_{cal}$. Roundels in (**d-e**) correspond to (**a-c**). Uncertainties in (**d-e**) are smaller than the data markers.

subresolution apertures show a different field dependence of $\rho_A$ that is similarly variable across lateral regions of only a few micrometers (Supplementary Fig. 9). This comparison shows that the field dependence of $\rho_A$ is a joint characteristic of the imaging system and emitter sample, and highlights the different information content of the different emitter arrays. Reference[4] reported a field dependence with a similar variability, although the authors did not explicitly discuss this aspect of their results, suggesting local effects of wavefront errors. Previous studies[5,6,13,14] have used particles to measure such errors, which are at least partially corrigible by calibration of the point spread function[4,8,41,42] or interpolation[50] such as in Ref. [4]. However, interpolant models include variation from all sources, including from particle size distributions, embedding defects in axial localization. Moreover, the need to sample the field still limits accuracy, such as near the periphery, where extrapolation or cropping may be necessary.

In a different analysis, we test Zernike polynomials[51] for widefield calibration. Classically, Zernike polynomials have modeled wavefront aberrations in phase space. Previously, we used Zernike polynomials to model aberration effects on apparent position in real space[8]. Presently, we show that Zernike polynomials can model aberration effects in real space, on both apparent lateral positions and image shapes. Our Zernike polynomials consist of a linear combination of the first 400 Noll indices, with prominent coefficients corresponding to astigmatism, defocus, and spherical aberration in phase space (Supplementary Fig. 10). In this way, Zernike polynomials can characterize aberrations in a way that empirical interpolation cannot. We presently compare the performance of Zernike polynomials, natural-neighbor interpolation[52], and nearest-neighbor interpolation[4] for widefield calibration.

**Localization accuracy throughout the focal volume.** To estimate the accuracy of widefield calibration, we take the values from the

local calibration functions $\{z(\rho_A)\}_{cal}$, $\{\Delta'_x(z)\}_{cal}$ and $\{\Delta'_y(z)\}_{cal}$ as the true values. The error at the location of each calibration particle is the residual of the fit of a Zernike model to these values, or, for interpolant models, the value of the interpolant after removal of the calibration particle. For a Zernike model, the errors (Fig. 5) shrink toward the corners of a square field (Supplementary Figs. 11 and 12) due to the rapid fluctuation of Zernike polynomials of high order near the periphery of the fitting domain. This corner effect potentially embeds errors in the calibration function, such as interpolation does, and results in a significant deviation of error histograms from normality, with a mean excess kurtosis of 0.66 ± 0.15. Errors from interpolation do not have this trend (Supplementary Figs. 13 and 14). However, our experimental particles are within a circular subset of the field (Figs. 1 and 6), resulting in error histograms that are closer to normal, with a mean excess kurtosis of 0.34 ± 0.17. These results suggest optimizing accuracy by calibrating an imaging field that encircles the sample by an increasing number of Zernike polynomials.

For $z$ values ranging from 2 μm above best focus to 2 μm below best focus, Zernike polynomials result in better accuracy than interpolation for both the square and circular fields (Fig. 5d–f), with minimum root-mean-square errors of 1.4 nm for $x$, 1.6 nm for $y$, and 62 nm for $z$. These values are the second of two main components of uncertainty in our study. We report uncertainties for widefield calibration as root-mean-square errors to include any non-zero mean value of error. This calibration reduces the systematic errors in lateral position (Figs. 2b–c and 4d–f) by a factor of two orders of magnitude, to within a root-mean-square error ranging from 1 nm to 5 nm (Fig. 5d–e). For all three widefield calibration models, axial localization errors are smaller above best focus than below best focus (Fig. 5f). Zernike polynomials model such field dependences, providing insight into the localization accuracy that is latent in intrinsic aberrations

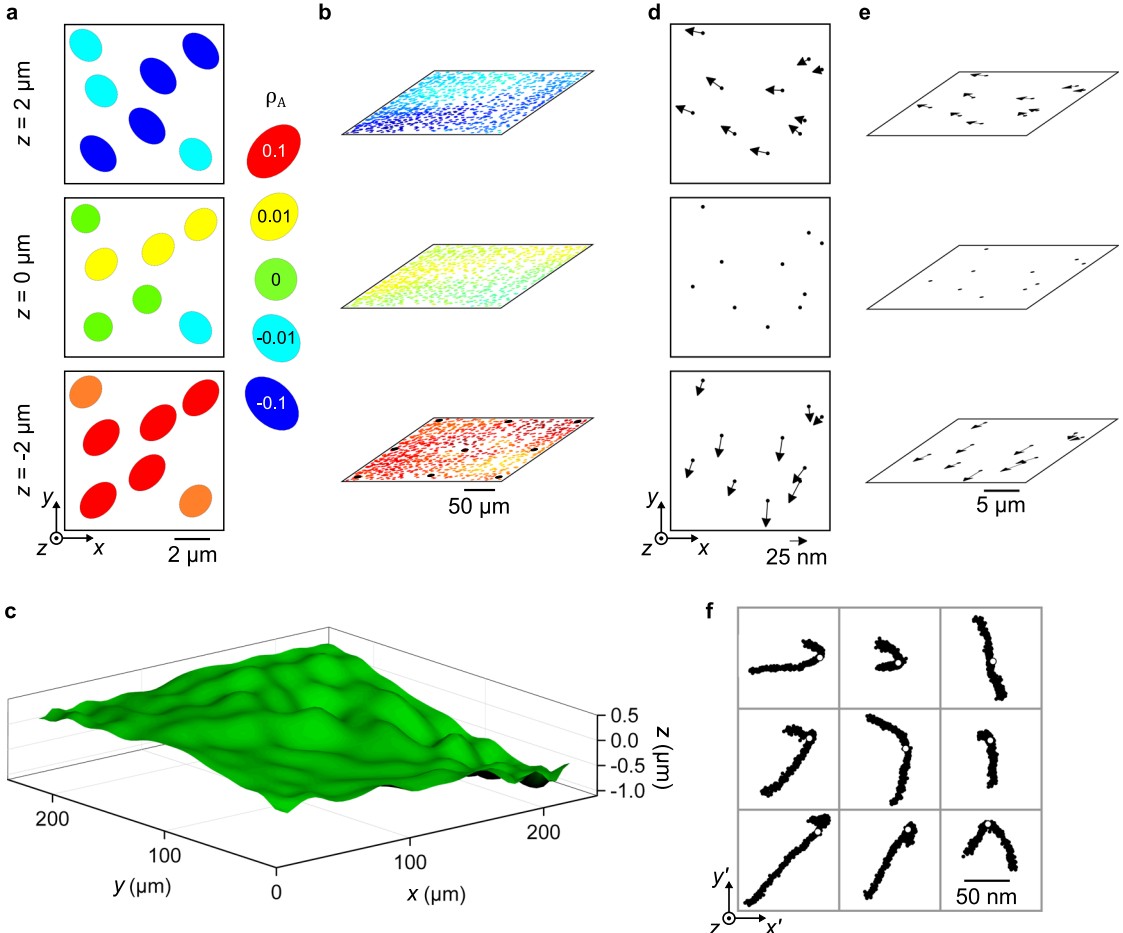

**Fig. 4 Field dependence of image shape and apparent lateral position. a** Schematic showing variation of image shape, with quantification by $\rho_A$, in three dimensions. **b** Scatter plots in perspective showing the lateral positions of the calibration particles and their values of $\rho_A$ for the three representative values of $z$ in (**a**). Black markers correspond to the particles in (**f**). **c** Surface plot showing a widefield calibration function of Zernike polynomials modeling variation in $z$ for a representative value of $\rho_A = 0$. **d**–**e** Vector plots showing the apparent lateral motion of a subset of calibration particles for the three representative values of $z$ in (**a**). **f** Grid of nine scatter plots showing the apparent lateral positions, through an axial range of 6 μm, of representative calibration particles from representative locations across the full lateral field. Black markers in the bottom plot of (**b**) show these representative locations. White data markers indicate the true lateral position of each particle, which we define as being at the $z$ position of best focus, $z_f$, for each particle.

throughout a deep and ultrawide field, and demonstrating the utility of our method to characterize optimal ranges of the focal volume.

We further test the accuracy of all three models by tracking the motion of a microscale body that moves through the focal volume (Fig. 6, Supplementary Note 2, Supplementary Fig. 15), validating the method and the preceding evaluation of uncertainty. Zernike models continue to outperform either method of interpolation, evidently due to the use of a model of optical aberrations, whereas empirical interpolation depends directly on data sampling. We subsequently use Zernike models as widefield calibration functions.

**Microsystem tracking**. We apply our measurement concept to track a complex microsystem (Fig. 6). In the process of localizing particles in three dimensions and using a rigid transformation to track motion in six degrees of freedom, the microsystem serves as a rotary microstage that enables rigorous tests of our method. For a particle constellation on a microscale body that moves with six degrees of freedom and rotates multiple times through the focal volume, periodic deviations from rigidity and planarity enable evaluation of the effects of the main components of uncertainty, as well as determination of the importance of field corrections for

distortion and apparent lateral motion that are possible to apply or omit (Supplementary Note 3).

The microsystem consists of a rotational electrostatic actuator coupling through a ratchet mechanism to a ring gear, forming a drive motor that operates in an open loop[34,35]. The ring gear has 200 teeth that couple to a load gear with 80 teeth and a diameter of 328 μm (Fig. 6a, b). A constellation of fluorescent particles rides on the load gear (Figs. 1 and 6). Each period of a square-wave voltage incrementally rotates the load gear, defining a quasi-static motion cycle, with 64 motion cycles completing a revolution. After each motion cycle, the microscope records a fluorescence micrograph of the particles on the load gear (Fig. 6c).

**Rigid transformations**. We track single particles on the load gear and fit rigid transformations in three dimensions to map particle positions between motion cycles. The center of rotation is a natural origin of our extrinsic coordinate system, which we determine as the mean value of all particle positions over all motion cycles. The residuals quantify the overall accuracy of the rigid transformations, with mean values of root-mean-square error of 2.0 nm in $x$, 2.1 nm in $y$, and 83 nm in $z$ (Supplementary Fig. 15). These values are consistent with the total uncertainty of localizing single particles (Supplementary Note 2), indicating

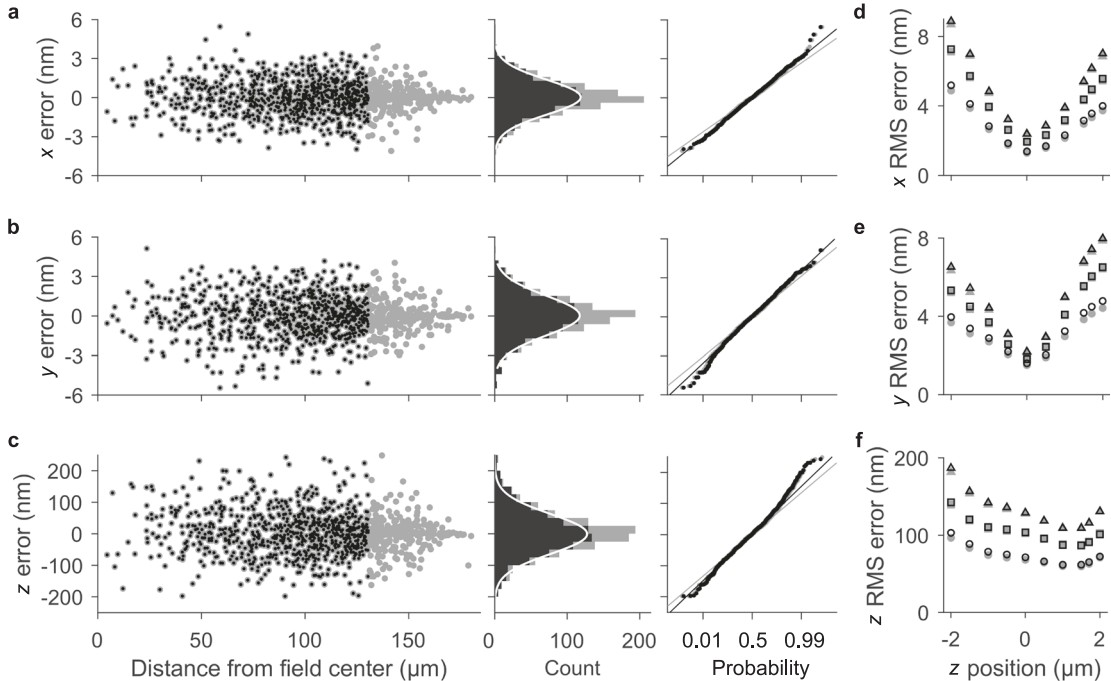

**Fig. 5 Localization error throughout a deep and ultrawide field.** Gray data include calibration particles from the full square field and black data include only the particles within a subset circular field. **a–c** Scatter plots, histograms, and normal probability plots of the differences between local and widefield calibration functions for each calibration particle, which define the error of widefield calibration, at the $z = 0$ focal surface for (**a**) $x$, (**b**) $y$, and (**c**) $z$. The scatter plots show these errors as a function of the distance of each particle from the nominal center of the field. Histograms include a (white line) Gaussian model fit to the black data. **d–f** Plots showing root-mean-square (RMS) error as a function of $z$ position for widefield calibration of (**d**) $x$, (**e**) $y$, and (**f**) $z$, using (triangle) nearest-neighbor interpolation, (square) natural-neighbor interpolation, and (circle) Zernike polynomials. Uncertainties in (**d–f**) are smaller than the data markers.

that the load gear is approximately rigid, but obscuring a slight deviation.

In the $z$ direction, the residuals of the rigid transformations fluctuate at a frequency of once per two revolutions with a relative amplitude of 6.6 % ± 0.5 % (Supplementary Fig. 15). The trajectories of single particles reveal a likely cause of this deviation (Supplementary Fig. 18), tracing a complex curvature of the top surface of the load gear. This curvature is approximately constant in space and time, indicating that the load gear flexes from its quasi-static coupling within the microsystem. Consistent with this result, for the 28 single particles under test on the load gear, the residuals of rigid transformations and the residuals of planar fits in the $z$ direction have a mean correlation of 0.20 with a standard deviation of 0.07. Moreover, Fourier analysis shows that both residuals share frequency content (Supplementary Fig. 18), including once per two revolutions. These results indicate that periodic flexure of the load gear, among other potential effects, causes deviation from rigidity in the $z$ direction. However, the effect is slight due to the incremental motion in comparison to the curvature from flexure, which has a low frequency in space.

This analysis demonstrates the complementary utility of localizing single emitters on a microscale body and fitting rigid transformations to the position data, even when the rigidity of the body is imperfect. Localization of single emitters in three dimensions yields a measurement of the underlying surface topography[53] and rigidity. Future work could improve this measurement by sampling at higher density, and using emitters with homogeneous sizes or fitting the localization data with a model that averages over random errors such as from a particle size distribution. Rigid transformation of the localization data is then a rich source of additional information. If systematic deviations from rigidity of the surface of the body are small in comparison to random errors from localization uncertainties of

single particles, then a rigid transformation meaningfully improves centroid and orientation precision[10] and enables tracking of additional degrees of freedom. Accordingly, we proceed with this characterization of the load gear as quasi-rigid.

**Microsystem motion in three dimensions**. A quasi-rigid body enables the combination of position information from multiple particles to track the centroid in three dimensions. The centroid trajectory shows a tilt of the load gear with respect to the imaging sensor (Fig. 6d), due to both tilt of the load gear relative to the substrate and tilt of the microsystem substrate relative to the imaging sensor. We subsequently refer to the plane of the imaging sensor, which is nearly parallel to the plane of the microsystem substrate, as *the* plane.

Position clusters in the $x$ and $y$ directions (Fig. 6d) result from ratcheting the load gear through 64 nominal orientations, causing each particle and the resulting centroid to revisit as many nominal locations with each revolution. Over 31 revolutions, the positions scatter due to play from clearances between parts of the microsystem, revealing imprecision of its intentional motion. Asymmetry of this scatter indicates a combination of translational play in the plane, which causes radial scatter about the center of each nominal location, and rotational play in the plane, which causes tangential scatter along the circular path of the centroid. Moreover, the variability of the axial position of the constellation centroid for all 64 nominal orientations of the load gear (Fig. 6d) indicates rotational play out of the plane, revealing unintentional motion of the microsystem.

Further effects of coupling interactions are apparent in the orientation $\hat{\theta}$ dependence of the ranges of the centroid motion in the radial $\hat{r}$ and axial $\hat{z}$ directions (Fig. 7). The range of radial motion is smallest in the direction of the coupling to the ring

**Fig. 6 Microsystem motion in three dimensions. a** Brightfield micrograph showing the drive motor, consisting of (i) a rotational actuator, (ii) a ring gear, and (iii) the load gear. **b** Brightfield micrograph magnifying the load gear and particles. The smaller dots with random spacing are florescent particles and the larger dots with regular spacing are etch holes. **c** Fluorescence micrograph showing a constellation of fluorescent particles on the surface of the load gear. The cross indicates the centroid of the subset of particles that we use for tracking. **d** Scatter plot showing the trajectory of the centroid of the particle constellation in three dimensions. Tilt is apparent. The position clusters in the *x* and *y* directions are due to the nature of the ratchet mechanism that rotates the load gear through 64 nominal orientations with each revolution. **e** Scatter plots showing centroid positions in the *x*-*y* plane for the nominal locations within the box in (**d**). Uncertainties for both lateral and axial positions are smaller than the data markers. **a** © 2020 IEEE. Reprinted, with permission, from Ref. [32].

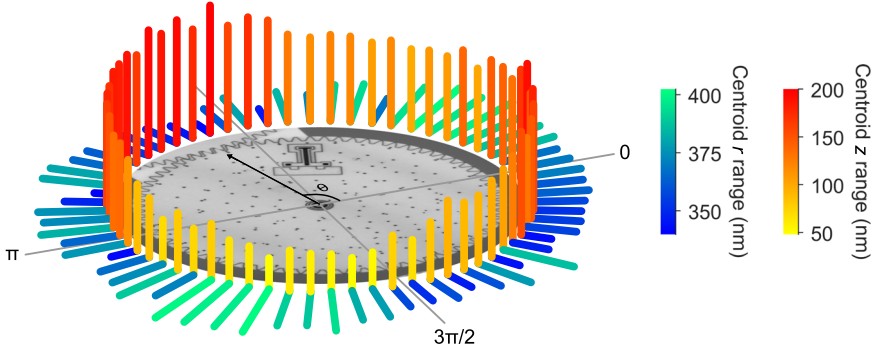

**Fig. 7 Play analysis.** Polar plot showing the range of motion of the constellation centroid in the radial direction and the axial direction at all 64 nominal orientations of the load gear. The bar length scales nonlinearly in the plane for clarity. The inset micrograph and black arrow show the direction of the coupling between the load gear and ring gear.

**Table 1 Motion variability due to mechanical play.**

| Degree of freedom | Mean value | Total range | Uncertainty components | | | Source of variability |
|---|---|---|---|---|---|---|
| | | | Localization | Drift | Total | |
| $\gamma$ (mrad) | 98.17 | 56.74 | 0.006 | – | 0.006 | Clearance between gear teeth, ring gear motion[9] |
| $\beta$ (mrad) | 9.1 | 20.1 | 3.4 | – | 3.4 | Clearance between load gear and substrate |
| $\alpha$ (mrad) | −593 | 3130 | 580 | – | 580 | Clearance between load gear and substrate |
| $\Delta_x$ (nm) | 0.6 | 787.1 | 0.4 | 2.49 | 2.52 | Clearance between load gear and hub |
| $\Delta_y$ (nm) | 0.3 | 774.3 | 0.4 | 1.29 | 1.36 | Clearance between load gear and hub |
| $\Delta_z$ (nm) | −0.02 | 140 | 17 | 12 | 21 | Uncertainty, clearance between load gear and substrate |

The term drift summarizes unintentional motion of the measurement system.
The uncertainty of image pixel size of 0.03 nm per pixel results in negligible errors for all values.

gear, which confines the translational play in the plane. The range of axial motion is smallest approximately $\pi/2$ rad from the coupling and approaches the uncertainty of $z$ for the centroid (Table 1), indicating vertical pinning of the load gear around this location. In contrast, the largest axial ranges occur around the coupling to the ring gear. A comparison with the centroid trajectory (Fig. 6d) shows that the coupling occurs at the top of the tilt and that the pinning occurs at the bottom of the tilt, indicating that the ring gear pins the load gear against either the substrate or an underlying part of the hub. This interaction is a probable cause of the flexure of the load gear, elucidating coupling interactions that cause unintentional motion of the microsystem and affect its characterization as a quasi-rigid body.

A vertical reciprocation in the trajectory of each particle occurs once per two revolutions (Supplementary Fig. 19). This frequency equals that of the fluctuations in the residuals of the rigid transformations in the $z$ direction (Supplementary Fig. 18), indicating a common cause of reciprocation and flexure. The total range in $z$ of the position of each particle at each nominal location is due in part to this reciprocation and is a result of the rotational play out of the plane. Accordingly, the maximum range in $z$ across all nominal locations of each particle increases with distance from the center of rotation (Supplementary Fig. 20). A linear fit gives the rotational play out of the plane by an arc-length approximation, with a slope of $9.38$ mrad $\pm 0.44$ mrad, which is consistent with the centroid trajectory (Fig. 6d). These results clarify and quantify the microsystem motion and further emphasize the utility of tracking both single and multiple particles on a quasi-rigid body.

**Microsystem motion in six degrees of freedom**. Realizing the full utility of our method, we measure the motion of the load gear in six degrees of freedom. The rigid transformations determine three translations $\Delta_x$, $\Delta_y$, and $\Delta_z$, and three rotations using a mixed coordinate system – the intrinsic rotation of the load gear $\gamma$ about the axis of rotation, the nutation $\beta$ of the axis of rotation with respect to the extrinsic $z$ axis, and the precession $\alpha$ of the axis of rotation about the extrinsic $z$ axis (Fig. 1). In two dimensions, uncertainties of motion measurements from rigid transformations are directly calculable from the positions and localization uncertainties of single particles[10]. An analytic extension to three dimensions is conceivable[54], but instead we evaluate the uncertainty of our motion measurements using Monte-Carlo simulations, propagating the experimental localization uncertainties of single particles through the rigid transformations (Supplementary Note 2).

Our measurements reveal significant variation in four of the six degrees of freedom due to play in the coupling of parts (Table 1, Fig. 8, Supplementary Fig. 19, Supplementary Movies 1, 2, and 3). The mean nutation $\bar{\beta}$ over each revolution (Fig. 8h) reciprocates with

a period of $4\pi$ radians, confirming that the vertical shift (Fig. 6d, Supplementary Fig. 19) occurs between sequential revolutions. The load gear translates little in the $z$ direction $\Delta_z$ (Fig. 8g), validating the corresponding uncertainty. Although the precession of the axis of rotation $\alpha$ varies over a wide range (Fig. 8c), the small nutation $\beta$ (Fig. 8b, d) causes the rigid transformations to be insensitive to this degree of freedom, so that most of the variability is within uncertainty. This is not a limitation of the method but is rather a consequence of the particular orientation of the load gear within the extrinsic reference frame of the imaging sensor. A different selection of reference frame could trade off these uncertainties against others. These results further elucidate our method and provide insights into the kinematics of complex microsystems.

In conclusion, we introduce the concept of fully exploiting the intrinsic aberrations of an optical microscope to accurately localize single emitters in three dimensions through a deep and ultrawide field. Our approach is counterintuitive, as the tendency is to consider intrinsic aberrations as defects to reduce through optical engineering, which increases the complexity and cost of optics, or to tolerate by error analysis, which quantifies the degradation of measurement performance. We invert this perspective to reveal and apply the latent capability of an ordinary microscope for axial localization, lowering the barrier to entry of localization microscopy in three dimensions. This result is important, because we also show that lateral accuracy generally requires axial localization. In this way, we elucidate and solve a fundamental problem of localization microscopy.

In the absence of optical engineering and in the presence of intrinsic aberrations, we develop a general and practical method for axial localization by Gaussian fitting. Several image parameters enable robust localization. For a constant emission intensity, an astigmatic defocus parameter yields useful precision and uniformity throughout a deep and ultrawide field. We elucidate the transition from local to widefield calibration, which is nonobvious due to the nonuniformity of the field of an ordinary microscope, even for field widths of less than ten wavelengths. We test several calibration functions to solve this fundamental problem, finding that Zernike polynomials model variations in astigmatic defocus and apparent position with the best accuracy, and characterize the utility of intrinsic aberrations of microscopes for our method.

In an application of our method, we introduce another concept of tracking emitters in three dimensions to measure the motion of a microscale body in six degrees of freedom. This analysis is analogous to tracking of point clouds at the macroscale, which is common and important. Comparisons and combinations of the trajectories of single emitters and rigid transformations in three dimensions elucidate both the tracking method and the microsystem motion in six degrees of freedom. Our method is immediately applicable to the imaging of fiducial particles for

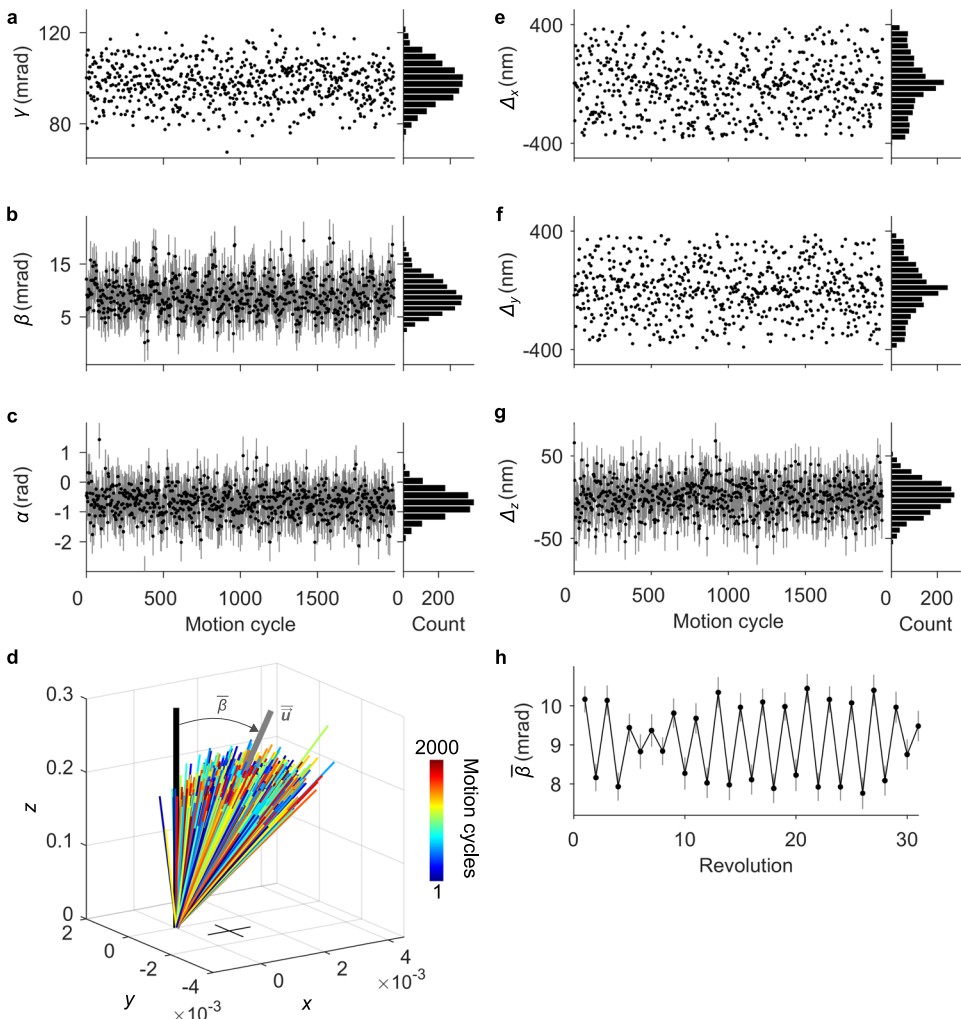

**Fig. 8 Microsystem motion in six degrees of freedom. a–c** Plots and histograms showing (**a**) intrinsic rotations of the load gear in three-dimensional space $\gamma$, (**b**) the angle between the axis of rotation and the extrinsic $z$ axis, or nutation $\beta$, and (**c**) the angle between the axis of rotation and the extrinsic $x$-$z$ plane, or precession $\alpha$. **d** Plot showing lines in the direction of (colors) the axis of rotation for each motion cycle, (gray) the mean axis of rotation, and (black) the extrinsic $z$ axis. The cross denotes uncertainties. **e–g** Plots and histograms showing translation of the load gear in the (**e**) $x$, (**f**) $y$, and (**g**) $z$ directions. **h** Plot showing the mean nutation $\bar{\beta}$ with a reciprocating rotation with each revolution of the load gear. Uncertainties are (**a**, **e**, **f**) smaller than data markers, (**h**) 68 % coverage intervals, and (**b**, **c**, **g**) as we describe in the Methods. **a–c**, **e–g** © 2020 IEEE. Reprinted, with permission, from Ref. [32].

the analytical leveling of imaging substrates, now in six degrees of freedom, complementing the analytical stabilization of instrument drift and characterization of aberration effects. As well, our method enables study of the motion of other microscale bodies.

We combine these methods of localization microscopy and rigid transformation and apply them to explore the motion of a complex microsystem. Our study reveals that nanoscale clearances between multiple parts in sliding contact not only degrade control of intentional motion but also cause unintentional motion in six degrees of freedom. Advancing practical measurements to study complex microsystems will help to fulfill their latent potential to perform reliably in applications that require multiradian rotations and other critical kinematics that are impossible to achieve by compliant mechanisms and impractical to measure by existing methods. Considering the importance of complex mechanical systems in the history of technology, it seems well worth the effort to understand and optimize their motion at small scales.

## Methods

**Optical microscope**. Our microscope has an ordinary combination of objective lens and tube lens, among other optics in their default configuration from the

manufacturer. The microscope has an inverted stand, a scanning stage that translates the sample in the $x$ and $y$ directions, and a piezoelectric actuator that translates the objective lens in the $z$ direction. The objective lens has air immersion, a working distance of 9.1 mm, a numerical aperture of 0.55, a nominal magnification of 50×, corrections for chromatic and flatness aberrations, and infinity correction. A light-emitting diode (LED) array and lens assembly yield nominal Köhler illumination of the sample. The emission spectrum of the LED is in Supplementary Fig. 21. The apochromatic tube lens has a focal length of 165 mm and a working distance of 60 mm. The tube lens focuses images onto a complementary metal-oxide-semiconductor (CMOS) camera with 2048 pixels by 2048 pixels, each with an on-chip size of 6.5 µm by 6.5 µm. The camera operates at a sensor temperature of −10 °C by thermoelectric and water cooling. We calibrate the microscope for these parameters[8]. The microscope records epifluorescence micrographs with a short-pass excitation filter with a transition at 628.0 nm, a dichroic mirror with a transition at 635.0 nm, and a long-pass emission filter with a transition at 634.5 nm. The microscope equilibrates for at least 1 h before use.

**Fluorescent samples**. We use polystyrene particles with a mean diameter of 1000 nm and a standard deviation of 24 nm containing boron-dipyrromethene fluorophores and having surface functionalization by carboxylic acid. We disperse the particles into pure water and microdeposit the suspension onto either a silicon substrate for calibration or the load gear of the microsystem for experiment. For magnification calibration, we fill an aperture array[1,8] with a solution of boron-dipyrromethene at a concentration of 750 µM in N,N-dimethylformamide. Emission spectra are in Supplementary Fig. 21.

**Calibration particles**. A scanning stage translates a random array of calibration particles on a silicon wafer across the lateral imaging field, acquiring micrographs through focus at each lateral position. The calibration and experimental particles are from the same population, but imaging occurs on different substrates – single-crystal silicon with a native oxide film for calibration, and polysilicon with a fluoropolymer film for the load gear. Evaluation of uncertainty indicates that any resulting difference of localization is insignificant. The data from all stage positions pool to calibrate the full field. A subset of particles forms images that differ significantly from the rest of the population of calibration particles, causing errors in the calibration data that are clearly systematic and not representative of the field dependence that we calibrate. Visual inspection confirms that these particles produce images with anomalous features, and we identify and cull such defective particles from the calibration data.

**Tilt correction**. In an initial analysis that is necessary to understand and calibrate axial dependences, we measure and correct any tilt of the calibration substrate relative to the $z$ axis by subtracting the plane of best fit from the surface of best focus[8]. This analytical leveling can replace the physical leveling of imaging substrates[8], which is rare even as samples extending across ultrawide lateral fields are becoming common. Moreover, the common use of fluorescent particles as fiducials for drift correction, and the application of our method, present the opportunity for a complementary correction of tilt.

**Microsystem imaging**. A combination of microsystem and microscopy parameters sets the imaging conditions. An exposure time of 1 ms reduces signal intensities of single particles to below the saturation threshold of the camera of 65,535 arbitrary units. The full lateral extent of the imaging field of 260 μm by 260 μm fits all experimental particles on the load gear within each micrograph, although the full diameter of the load gear still exceeds this lateral extent. The rotation of the load gear moves the particles so far as to require operation of the camera using a global shutter. In this mode, the camera triggers the LED illumination *on* when the entire sensor is exposing, rather than a rolling shutter for which pairs of pixel rows expose sequentially. A global shutter eliminates motion artifacts from a rolling shutter but introduces a delay between sequential images due to the readout time. For a global shutter, the imaging frequency is $1/(\tau_e + (n_{prp} \times 10 \text{ μs}))$, where $\tau_e$ is the exposure time and $n_{prp}$ is the number of pixel row pairs, which defines the readout extent and has a maximum value of 1024. A decrease of readout extent enables imaging frequencies extending into the kilohertz range[9]. There is a trade-off, however, as a decrease of readout extent also decreases the total number of signal photons from multiple emitters that contribute precision to a rigid transformation (Supplementary Fig. 22). In our measurements, the readout time of 10 ms for the full sensor, in combination with the exposure time of 1 ms, permit a maximum imaging frequency of 90.9 Hz ± 0.2 Hz to sample the quasi-static motion of the microsystem as rapidly as possible while any unintentional motion of the measurement system is ongoing. This imaging frequency sets the frequency of a square wave voltage with both an amplitude and offset of approximately 7.5 V, driving the rotational actuator that couples to the load gear. Synchronization of micrograph acquisition to the end of each period of the square wave allows sufficient time for the microsystem to settle into a static state, avoiding imaging artifacts.

**Gaussian fitting**. Light-weighting[8] fits bivariate Gaussian models to emitter images and extracts parameters.

**Polynomial models**. Empirical polynomial functions of order 16 model the $z$ dependence of Gaussian parameters (Fig. 2b, c, e, f, Supplementary Fig. 4), and order 5 to determine $z_f$ (Fig. 2d). This polynomial order yields approximately normal residuals of corresponding fits to data. Inversion of the functions for image shape provides a local calibration function with $z$ position as the dependent variable (Supplementary Table 2). We fit polynomial models to data and calculate the coefficients of Zernike models using least-squares estimation with uniform weighting and the Levenberg–Marquardt algorithm.

**Rigid transformations**. The iterative-closest-point algorithm[55] determines rigid transformations $T_i$ that map particle positions $\vec{p}_i$ between consecutive motion cycles $i$ and $i-1$,

$$\vec{p}_i = T_i \vec{p}_{i-1} \tag{7}$$

where

$$T_i = \begin{pmatrix} r_{11} & r_{12} & r_{13} & 0 \\ r_{21} & r_{22} & r_{23} & 0 \\ r_{31} & r_{32} & r_{33} & 0 \\ \Delta_x & \Delta_y & \Delta_z & 1 \end{pmatrix}_i \tag{8}$$

and

$$\vec{p}_i = \begin{pmatrix} \{x_1 \dots x_j\} \\ \{y_1 \dots y_j\} \\ \{z_1 \dots z_j\} \end{pmatrix}_i \tag{9}$$

for $j$ particles.

The axis-angle representation describes the rotations of the load gear. The direction of the eigenvector of the rotation matrix

$$R_i = \begin{pmatrix} r_{11} & r_{12} & r_{13} \\ r_{21} & r_{22} & r_{23} \\ r_{31} & r_{32} & r_{33} \end{pmatrix}_i \tag{10}$$

with corresponding eigenvalue 1,

$$\vec{u}_i = \begin{pmatrix} r_{32} - r_{23} \\ r_{13} - r_{31} \\ r_{21} - r_{12} \end{pmatrix}_i = \begin{pmatrix} u_1 \\ u_2 \\ u_3 \end{pmatrix}_i \tag{11}$$

determines the axis of rotation in our extrinsic coordinate system, and the magnitude

$$|\vec{u}| = 2 \sin(\gamma) \tag{12}$$

determines the intrinsic rotation $\gamma$ of the load gear about that axis. The two additional degrees of freedom are the nutation

$$\beta = \frac{\pi}{2} - \sin^{-1}\left(\frac{u_3}{|\vec{u}|}\right) \tag{13}$$

and the precession

$$\alpha = \text{atan2}(u_2, u_1) = \tan^{-1}\left(\frac{u_2}{u_1}\right) + \frac{\pi}{2}\text{sgn}(u_2)(1 - \text{sgn}(u_1)) \tag{14}$$

## Data availability

The data supporting the findings of this study are available from the corresponding author upon reasonable request.

## Code availability

The code and sample data supporting the findings of this study are available as Supplementary Software at https://doi.org/10.18434/mds2-2376.

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

## Acknowledgements

The authors acknowledge Christopher Wallin and J. Alexander Liddle for insightful reviews and helpful comments, Glenn Holland for technical support, and support of this research under the NIST Innovations in Measurement Science program. C.R.C. acknowledges support of this research under the Cooperative Research Agreement between the University of Maryland and the National Institute of Standards and Technology Center for Nanoscale Science and Technology, Award 70NANB10H193, through the University of Maryland.

## Author contributions

S.M.S. supervised the study. J.G. and S.M.S. conceived the experiments. C.R.C. performed the experiments with contributions from S.M.S. C.D.M. and J.G. conceived an initial analysis of localization data in two dimensions and three degrees of freedom. C.D.M. and C.R.C. performed the initial analysis with contributions from J.G. C.R.C. and S.M.S. conceived the final analysis of localization data in three dimensions and six degrees of freedom, involving intrinsic aberrations and field dependences. C.R.C. performed the final analysis with contributions from S.M.S. C.R.C. and S.M.S. interpreted the microsystem motion with contributions from J.G. and C.D.M. B.R.I. fabricated the aperture array. C.R.C. and S.M.S. wrote the manuscript with contributions from J.G. and C.D.M.

## Competing interests

The authors declare no competing interests.
