## [Peer Review File · Nature Communications]

REVIEWER COMMENTS

Reviewer #1 (Remarks to the Author):

The main claim of the paper is a technique, which requires aberrations in the optics to localize the center of emitters in six degrees of freedom. The idea is realized and the measurement of 3D MEMS motion is demonstrated. It is a well-written understandable paper.

Comments:

1) In the paper it is not defined what emitters are used in the experiments. Sometimes they are called particles. Are these for example fluorescence markers or gold-nano-beads? The authors should clarify this important part of their concept. The concept requires point emitters with a diameter substantially below the wavelength of the measurement-laser light in order to have a defined aberration in the image of the emitters if they are out-of-focus. The aberration is correlated to the z-displacement. Since the emitters are that important, the emitters have to be discussed in detail. How is the contrast in respect to the light reflected at the structure achieved?

2) Localization Microscopy like PALM has the goal to enhance the resolution of single images. I wonder how the statistically distributed emitters could help to enhance the resolution of the 3D-shape of a MEMS device, because the emitters are not correlated to the device structure, they are randomly distributed on the surface and their exact positions in respect to the structure geometry is unknown. The example shown in the paper demonstrates the measurement of movements and overall positions of a solid structure (6 degrees of freedom not of a single emitter but of a cluster of emitters fixed on the specimen) and here the use of bright emitters with unknown positions make sense. It seems that the displacement in 6 degrees of freedom is more important than the localization of the emitters for the chosen example. The reference 31 given for the importance to measure 3D motions is indeed a technique sensitive only to displacements and not to the specimen shape. Thus, I think that localization of the emitters is not that important for the chosen example. The coordinates of the localized emitters are used to estimate the six-degree-of-freedom displacement of a solid structure with very high resolution. This seems to be the important achievement in the submitted paper.

3) This third comment is correlated to comment 2. Since the displacement information seems to be the important information, I wonder why the authors give very little inside in the 3D-motion measurement of MEMS. Displacement of MEMS surfaces can also be extracted from 3D data of altering-focus microscopes, coherence scanning interferometers, phase-shifting-interferometers or holographic microscopes with in-plane displacement resolutions much below the Abbe resolution limit. Therefore, it would be important to compare the improvement of the localization technique to this state of the art. Early work of 3D motion measurements was accomplished by research groups of Freeman (<https://doi.org/10.1557/mrs2001.66>), Muller (<https://doi.org/10.1109/JMEMS.2002.803285>), Bossebeboeuf ([https://doi.org/10.1016/S0143-8166\(01\)00040-9](https://doi.org/10.1016/S0143-8166(01)00040-9)) or Depeursinge (<https://doi.org/10.1364/OE.15.007231>). These techniques achieve nanometer or even subnanometer resolutions by "fitting techniques" and averaging. In the submitted paper utilizes also "fitting technique" to localize the centers of circular emitters with known aberrations of the microscopy and known deformations of the PSF. Motions are calculated by subtracting the positions of the emitters. I think, the paper should highlight more the results in this context because the localization of emitters distributed statistically on the

surface itself seems not to be the final attempt. It just links a random accurate x,y coordinate uncorrelated to the structure geometry with an accurate z-coordinate. Standard interferometry links a little-bit inaccurate known x,y coordinate correlated to the structure geometry with an accurate z-coordinate. However, I would expect similar accuracy for 3D displacements. Therefore, I would expect a discussion in the paper.

4) The measurement frequency bandwidth is very important for motion measurement systems. The authors do not give any information about the bandwidth performance of the presented technique. Have the authors combined there technique with strobe technique to achieve high bandwidths? This is not a requirement for publication but it should be discussed.

5) I think, the paper is too long. The part of the localization procedure should be shortened. It is quite understandable how the localization is achieved with the calibrated intrinsic aberration. However, I cannot see clearly the advantage for shape or motion measurements in MEMS. Therefore, the authors should improve this part of the paper. Is the localization of the emitters valuable for shape measurements? The x,y-localization is more accurate than the measurement with an microscope interferometer but the positions of the emitters are random. Thus, what is the improvement in shape measurements? The exact knowledge of the position does not matter for motion measurements, which is the demonstrated example. However, what are the improvements compared with other techniques? Reference 31 shows resolutions of displacements in the picometer regime for MHz frequencies. The presented technique may have an advantage at tilt measurements. I mentioned prior work in 3). How does the new technique compare to the results presented in these papers and reference 31. In addition, commercially available systems from Bruker, Polytec or Lynceetec do provide time-dependent 3D shape data and these well-optimized techniques present the state of the art of the techniques introduced in above-mentioned papers. How, does the presented technique compare to these commercially available systems? How does it improve the state of the art of MEMS testing in terms of parameters like resolution and uncertainty of displacements (or vibration amplitudes in the spectrum)?

I recommend to revise the paper in respect to these comments.

Reviewer #2 (Remarks to the Author):

In the article "Accurate localization microscopy in six degrees of freedom by intrinsic aberration calibration," the authors present an interesting and novel method for accurately locating point scatterers in three dimensions from a single two-dimensional image. The authors use the presence of aberrations to identify the particles' axial positions, in a similar vein to approaches that add astigmatism to the optical train to encode axial positions in the recorded image. The authors identify the axial position of particles by first imaging a series of calibration particles and fitting bi-variate Gaussians to the data. They then fit the extracted parameters of the Gaussian to continuous functions of the particles' axial positions. The authors then fit new images of new particles to Gaussians and use the fitted parameters to calculate the actual axial position of the new particle. The authors close by demonstrating their technique on particles stuck to a rotating gear, convincingly showing that the gear flexes as it rotates.

Using aberrations intrinsic to the objective lens to track particles in three dimensions is a novel technique that warrants publication. In addition, the authors conclusively demonstrate that typical particle localization algorithms can have strong biases and inaccuracies when

aberrations are ignored. These inaccuracies can be significantly larger than the limit of precision. In the manuscript, the authors then show how to correct for these localization inaccuracies due to aberrations. This work provides both novel techniques and raises interesting questions for researchers using localization microscopy.

However, while the authors have presented an interesting approach for extracting 3D information from a 2D image by leveraging intrinsic aberrations in the objective lens, I feel that the paper needs more validations of the technique before it is published. The plots and discussions of uncertainties in the paper (e.g. Fig. 2, 3, and 5) all show the uncertainties for the data used to fit the models. The authors should measure the accuracy of their technique on validation data that is distinct from the calibration data. Including an empirical measurement of uncertainties from the imaged gear as validation data would suffice. The authors could examine the residuals of the fit to the rigid translations of the gear, using the fact that the true displacements of the particles are rigid body rotations (or rigid-body rotations with flexure). Properties of these residuals are mentioned in the text, on pg 17, but the actual value of the measurement errors are not mentioned, and there is no figure that shows their detailed properties.

This validation is especially important because the authors have made several choices in developing their algorithm that could use some justification. The authors fit images of the emitters to Gaussians, although real optical point-spread functions are not Gaussian, even in the absence of aberrations. The authors use particles which are not sub-resolution, and therefore the results may change as the particle size changes. And the authors make some choices with normalization of parameters for estimating the particle axial position (i.e. the choice of normalization for ρ_w and ρ_A) that could give results that change as the particles photobleach. Adding a detailed analysis of the algorithm's performance on the validation data would assuage any concerns about the importance of these design choices when applying the algorithm to new data. I would prefer to see accuracies measured on validation data over details about the kurtosis of the fit residuals or the rotation of the gear.

In addition to this main concern, I have several comments of lesser importance:

* The authors should be more clear about the size of the particles used. In the caption to Fig. 1, the authors say that the particles are 1 micrometer. However, they do not say this in the main text, and they do not say what size the emitters are for the aperture arrays or the gear. This is important, as variation in particle sizes may change both fitted amplitude (via changes in the emission intensity) and the fitted Gaussian widths (as the particles are not sub-resolution).

* The discussion on pg 10 regarding the fitting of the calibration functions is confusing. The authors first state that they fit $z(\rho_a)$, then $x'(z)$, then $z(x', y', \rho_a)$. Are the authors fitting an axial displacement which depends on both the in-plane position and the normalized correlation? Or are the authors just fitting the axial displacement to the normalized correlation?

* On pg 7, the authors state that their bivariate Gaussian "approximates the image loci as ellipses with axes at an angle of $\pi/4$ radians with respect to the x and y axes." This is only true if w_x is restricted to be equal to w_y . The functional form used in the equation on pg 5 provides a full description of the covariance matrix and thus allows the axis to be oriented at an arbitrary angle to the x- or y- axes.

* The "astigmatic defocus parameter" $\rho_w = \rho * (|w_x| + |w_y|) / 2$ is not geometrically invariant. For a system with astigmatism extrinsically added in, it makes sense to use a defocus parameter specific to a coordinate frame, since the astigmatism axis is known. For this system without a preferred coordinate system, it might be better to use a geometrically invariant measure. This is a detail, and not especially important, but it is choices like this that make me want a clearer demonstration on a validation set outside of the calibration data.

* The authors state that they use a 16-order polynomial to model the z-dependence of the Gaussian parameters, and a 5th-order to choose z_f . A sentence explaining why these orders were chosen might be helpful.

REVIEWER COMMENTS

We sincerely thank both reviewers for their time, consideration, and insights. We have carefully considered all of their comments. To address them, we have performed new analyses and made major revisions to, and clarifications of, our manuscript. In so doing, we believe that we have significantly improved our work, and completely addressed the comments of the reviewers.

Reviewer #1 (Remarks to the Author):

The main claim of the paper is a technique, which requires aberrations in the optics to localize the center of emitters in six degrees of freedom. The idea is realized and the measurement of 3D MEMS motion is demonstrated. It is a well-written understandable paper.

We thank the reviewer for these positive comments. We would describe our main claims as follows. First and foremost is the new concept of fully exploiting the latent information of intrinsic aberrations by complete calibration of an ordinary microscope, enabling accurate localization of single emitters in three dimensions across an ultrawide and deep field. This is an important advance of general interest that extends beyond any specific application of the method. Our application to measure microsystem motion in six degrees of freedom is another new concept and important advance for microscale motion metrology, while also providing a rigorous test that validates the tracking method.

Comments:

1) In the paper it is not defined what emitters are used in the experiments. Sometimes they are called particles. Are these for example fluorescence markers or gold-nano-beads? The authors should clarify this important part of their concept.

We respectfully note that the submitted manuscript presented the information that the reviewer requested in the Supplementary Methods section titled *Fluorescent Samples*, including the particle size distribution, material composition, surface functionalization, chemical composition of sorbed fluorescent dye, and the spectral properties of the measurement system (Supplementary Figure S20). To more clearly present this information, we have revised the manuscript so that this information appears in the Methods section of the main text. We state that we use fluorescent particles at several points in the main text, such as the caption for Figure 1d and the caption for Figure 6. We use the term “emitters” when we are speaking more generally about localization microscopy, as our methods are not limited to particles as the type of emitter. We have carefully reviewed our usage of these two terms, and we believe that each usage is as we intend it.

The concept requires point emitters with a diameter substantially below the wavelength of the measurement-laser light in order to have a defined aberration in the image of the emitters if they are out-of focus. The aberration is correlated to the z-displacement. Since the emitters are that important, the emitters have to be discussed in detail. How is the contrast in respect to the light reflected at the structure achieved?

We respectfully note that emitters with sizes that are similar to or larger than the imaging wavelength can have images that are meaningfully aberrated out of focus. In our study, the fluorescent particles have a mean diameter of 1 μm and thus are above the resolution limit set by the peak emission wavelength of approximately 650 nm (Supplementary Figure S20), and yet the particle images show significant effects of the intrinsic aberrations of the imaging system, enabling our new method of axial tracking. The contrast mechanism in our study is fluorescence, allowing selective imaging of fluorescence emission from microparticles and spectral rejection of excitation wavelengths reflected from the microsystem surface. Some emission light from the particle is likely reflected by the device surface and collected by the imaging system, and we use a similar silicon substrate for the

calibration particles to account for this effect. We have revised the manuscript to emphasize the importance of using the same population of emitters for both calibration and experiment.

2) Localization Microscopy like PALM has the goal to enhance the resolution of single images. I wonder how the statistically distributed emitters could help to enhance the resolution of the 3D-shape of a MEMS device, because the emitters are not correlated to the device structure, they are randomly distributed on the surface and their exact positions in respect to the structure geometry is unknown.

We thank the reviewer for this insightful comment. We have revised our manuscript to include a new discussion of this interesting capability of localization microscopy. We have also added a new reference (52) to a recent publication in *Science Advances*, which focuses on surface profilometry by localization microscopy in three dimensions, applying the standard paradigm of engineering the point spread function by a cylindrical lens. In our study, rigid transformation of localization data in three dimensions yields not only measurements of microsystem motion in six degrees of freedom, but also reveals flexure of the load gear during operation of the microsystem. We reduce this aspect of the data, which corresponds to a change in surface topography, as a deviation from planarity. A different analysis of the localization data, such as by polynomial modeling, could yield further details of surface topography. However, our new method, as we present it in this study, is not yet optimal for such analysis. Future work could improve this aspect of the measurement by sampling the surface with a higher density of emitters, using emitters with homogeneous sizes such as single molecules, or optimizing the polynomial model to capture surface curvature and average over errors such as from a particle size distribution. We think, and reference (52) supports, that further details of surface profilometry are beyond the scope of the present study. In comparison to the new reference (52), our work has a different focus and develops different methods to quantify different measurands, yielding a complete article near the length limit of the journal, as well as a supplement of significant length.

For the reviewer, we note that it is possible to correlate emitter positions to device structure by acquiring micrographs in both fluorescence and brightfield imaging modes (Figure 5b-c). This imaging procedure can be performed on a static device. Or, in the case of quasi-static operation, such as in our study, an additional brightfield micrograph can be recorded during operation of the device, in sequence with each fluorescence image and with some loss of bandwidth. The particle locations in the fluorescence images can be accurately correlated with the brightfield image of the device structure by a calibration and localization of brightfield images of device structures. As above, we think that such measurements are interesting and beyond the scope of the present study.

The example shown in the paper demonstrates the measurement of movements and overall positions of a solid structure (6 six degree of freedoms not of a single emitter but of a cluster of emitters fixed on the specimen) and here the use of bright emitters with unknown positions make sense. It seems that the displacement in 6 degrees of freedom is more important than the localization of the emitters for the chosen example. The reference 31 given for the importance to measure 3D motions is indeed a technique sensitive only to displacements and not to the specimen shape. Thus, I think that localization of the emitters is not that important for the chosen example. The coordinates of the localized emitters are used to estimate the six-degree-of freedom displacement of a solid structure with very high resolution. This seems to be the important achievement in the submitted paper.

The reviewer is correct that our application of the method to measuring MEMS motion was with the intent of measuring the motion of the device in six degrees of freedom, during single motion cycles. We are not certain, but the reviewer might mean to comment that the displacements of the emitters are more important than the locations of the emitters relative to the device surface. This is true for our measurements of motion. The *localization* of the emitters to determine their relative positions between motion cycles, on the other hand, is a core aspect of our method. As in our response to the

reviewer's summary of the work, we agree that the measurement of MEMS motion in six degrees of freedom is an important advance. Moreover, this measurement is based on new and generally applicable concepts in localization microscopy, which we think are of even greater importance to a general audience with a broad variety of applications of interest.

3) This third comment is correlated to comment 2. Since the displacement information seems to be the important information, I wonder why the authors give very little insides in the 3D-motion' measurement of MEMS. Displacement of MEMS surfaces can also be extracted from 3D data of altering-focus microscopes, coherence scanning interferometers, phase-shifting- interferometers or holographic microscopes with in-plane displacement resolutions much below the Abbe resolution limit. Therefore, it would be important to compare the improvement of the localization technique to this state of the art. Early work of 3D motion measurements was accomplished by research groups of Freeman (<https://doi.org/10.1557/mrs2001.66>), Muller (<https://doi.org/10.1109/JMEMS.2002.803285>), Bossebeboeuf ([https://doi.org/10.1016/S0143-8166\(01\)00040-9](https://doi.org/10.1016/S0143-8166(01)00040-9)) or Depeursinge (<https://doi.org/10.1364/OE.15.007231>). These techniques achieve nanometer or even subnanometer resolutions by "fitting techniques" and averaging. In the submitted paper utilizes also "fitting technique" to localize the centers of circular emitters with known aberrations of the microscopy and known deformations of the PSF. Motions are calculated by subtracting the positions of the emitters. I think, the paper should highlight more the results in this context because the localization of emitters distributed statistically on the surface itself seems not to be the final attempt. It just links a random accurate x,y coordinate uncorrelated to the structure geometry with an accurate z-coordinate. Standard interferometry links a little-bit inaccurate known x,y coordinate correlated to the structure geometry with an accurate z-coordinate. However, I would expect similar accuracy for 3D displacements. Therefore, I would expect a discussion in the paper.

The expectation of the reviewer is reasonable. We have revised our manuscript and supplement to more clearly distinguish our new method from other microscopy and interferometry methods to measure micromechanical motion and surface topography. We respectfully note that we have made these revisions in consideration of both the length limitation of *Nature Communications*, and our purpose of publishing an original research article rather than a review article. In brief summary, we are able to directly measure micromechanical motion in six degrees of freedom for the first time, and we are able to do so with a simple and economical measurement system consisting of an ordinary optical microscope. We emphasize that both the new capability and broad accessibility of our measurement method are critical for the general audience of *Nature Communications*. We present a concise discussion in the introduction that includes several new references and refers to a new Supplementary Table S1. This table sets out relevant metrics at an appropriate level of detail for an original research article. We think that these revisions are the appropriate way to address this comment without distraction.

4) The measurement frequency bandwidth is very important for motion measurement systems. The authors do not give any information about the bandwidth performance of the presented technique. Have the authors combined there technique with strobe technique to achieve high bandwidths? This is not a requirement for publication but it should be discussed.

We agree with this important comment from the reviewer. We have revised the manuscript to include a new description of the measurement frequency in the Methods, as well as a new Supplementary Figure S21 that describes the tunability of CMOS imaging sensors by varying exposure time and readout extent, in the context of our measurements. For measuring consecutive single motion cycles in series, measurement frequency is limited by the acquisition frequency of the imaging sensor. In general, a higher frequency requires a shorter exposure time for each optical micrograph, which decreases the number of photons that we collect from each emitter, reducing localization precision. Thus, there is a tradeoff between the frequency and precision of motion tracking. The imaging system in this study consists of a powerful LED for widefield illumination of intensely fluorescent

particles, allowing subnanometer lateral precision at an imaging frequency of approximately 90 Hz. In our specific case of using a CMOS imaging sensor, there is a characteristic tradeoff between imaging frequency and field size. We have previously demonstrated how this capability of CMOS sensors enables imaging and measurement at a frequency of 1 kHz by reducing the readout extent of imaging field (<https://doi.org/10.1109/JMEMS.2018.2874771>). The extension from measurements in two dimensions to three dimensions does not affect bandwidth. The reviewer asks a prescient question about stroboscopy – we have indeed combined our method of tracking with stroboscopic illumination to achieve even higher bandwidths. We are currently preparing the results of that study for a future publication.

5) I think, the paper is too long. The part of the localization procedure should be shortened. It is quite understandable how the localization is achieved with the calibrated intrinsic aberration.

We respectfully note that our submitted manuscript was, and our revised manuscript still is, within the length limits for *Nature Communications*. Our revised manuscript is approximately 58 % microscopy and 42 % MEMS motion, which we think is appropriate for the interests of the readership of the journal. The general advances that our method yields for localization microscopy are likely to be of proportionally greater interest to a general audience, such as that of *Nature Communications*, than are the advances in the important but specific application to measure MEMS motion. In particular, the lateral field dependence of the axial dependence of both image shape and apparent lateral motion is critical for accurate measurements and nontrivial to describe with appropriate detail. The current length of the main text, and significant supplementary material, together show the many considerations and details necessary for a comprehensive study of this topic.

However, I cannot see clearly the advantage for shape or motion measurements in MEMS. Therefore, the authors should improve this part of the paper. Is the localization of the emitters valuable for shape measurements? The x,y-localization is more accurate than the measurement with an microscope interferometer but the positions of the emitters are random. Thus, what is the improvement in shape measurements?

This comment seems to reiterate the reviewer's previous comments, and so we refer the reviewer to our responses to those comments while noting that our work presents significant advances in both localization microscopy and microscale motion metrology, irrespective of surface topography and shape measurements. We have clarified that we have introduced shape measurements in three dimensions from intrinsic aberrations without having sought to optimize these measurements.

The exact knowledge of the position does not matter for motion measurements, which is the demonstrated example. However, what are the improvements compared with other techniques? Reference 31 shows resolutions of displacements in the picometer regime for MHz frequencies. The presented technique may have an advantage at tilt measurements. I mentioned prior work in 3). How does the new technique compare to the results presented in these papers and reference 31. In addition, commercially available systems from Bruker, Polytec or Lynceetec do provide time-dependent 3D shape data and these well-optimized techniques present the state of the art of the techniques introduced in above-mentioned papers. How, does the presented technique compare to these commercially available systems? How does it improve the state of the art of MEMS testing in terms of parameters like resolution and uncertainty of displacements (or vibration amplitudes in the spectrum)?

We refer the reviewer to our response to comment 3 for how we have addressed this comment.

I recommend to revise the paper in respect to these comments.

Reviewer #2 (Remarks to the Author):

In the article "Accurate localization microscopy in six degrees of freedom by intrinsic aberration calibration," the authors present an interesting and novel method for accurately locating point scatterers in three dimensions from a single two-dimensional image. The authors use the presence of aberrations to identify the particles' axial positions, in a similar vein to approaches that add astigmatism to the optical train to encode axial positions in the recorded image. The authors identify the axial position of particles by first imaging a series of calibration particles and fitting bi-variate Gaussians to the data. They then fit the extracted parameters of the Gaussian to continuous functions of the particles' axial positions. The authors then fit new images of new particles to Gaussians and use the fitted parameters to calculate the actual axial position of the new particle. The authors close by demonstrating their technique on particles stuck to a rotating gear, convincingly showing that the gear flexes as it rotates.

Using aberrations intrinsic to the objective lens to track particles in three dimensions is a novel technique that warrants publication. In addition, the authors conclusively demonstrate that typical particle localization algorithms can have strong biases and inaccuracies when aberrations are ignored. These inaccuracies can be significantly larger than the limit of precision. In the manuscript, the authors then show how to correct for these localization inaccuracies due to aberrations. This work provides both novel techniques and raises interesting questions for researchers using localization microscopy.

We thank the reviewer for this complementary assessment of our work.

However, while the authors have presented an interesting approach for extracting 3D information from a 2D image by leveraging intrinsic aberrations in the objective lens, I feel that the paper needs more validations of the technique before it is published. The plots and discussions of uncertainties in the paper (e.g. Fig. 2, 3, and 5) all show the uncertainties for the data used to fit the models. The authors should measure the accuracy of their technique on validation data that is distinct from the calibration data. Including an empirical measurement of uncertainties from the imaged gear as validation data would suffice. The authors could examine the residuals of the fit to the rigid translations of the gear, using the fact that the true displacements of the particles are rigid body rotations (or rigid-body rotations with flexure). Properties of these residuals are mentioned in the text, on pg 17, but the actual value of the measurement errors are not mentioned, and there is no figure that shows their detailed properties.

We agree with the reviewer that tracking the quasi-rigid-body motion of the particles on the MEMS gear as they rotate throughout the ultrawide and deep imaging field is an appropriate test of our method. We refer the reviewer to Supplementary Note 2, Supplementary Figures S14, S15, S16, and S17, and Supplementary Table S3, in which we show and describe in detail this type of validation. Indeed, we find that the transformation errors are in good agreement with the uncertainties that we expect from our localization method, which we quantify in Figures 2, 3, and 5.

Moreover, we perform Monte-Carlo simulations that propagate the localization uncertainties that we empirically determine from the residuals of the rigid transforms through the rigid transform model to determine the uncertainty of motion measurements for each degree of freedom. We describe this procedure in detail in Supplementary Note 2 and we have moved the table containing the resulting values from the Supplementary Information to the main text.

This validation is especially important because the authors have made several choices in developing their algorithm that could use some justification. The authors fit images of the emitters to Gaussians, although real optical point-spread functions are not Gaussian, even in the absence of aberrations.

The authors use particles which are not sub-resolution, and therefore the results may change as the particle size changes.

While the reviewer is correct that the true optical point-spread functions are not exactly modeled by Gaussian functions, the particles we use in this work are larger than the resolution limit, having a nominal diameter of 1 micrometer, and produce images that are approximately Gaussian. As we demonstrate, a bivariate Gaussian is indeed a useful approximation for such images and performs well in capturing the shape of the particle images, both for lateral and axial localization. In addition, the calibration procedure accounts for any localization errors that result from this useful but inexact model approximation. We agree that the calibration of image shape will change with particle size, and we have made revisions to further emphasize the importance of using particles from the same population for both calibration and experiment.

And the authors make some choices with normalization of parameters for estimating the particle axial position (i.e. the choice of normalization for ρ_w and ρ_A) that could give results that change as the particles photobleach. Adding a detailed analysis of the algorithm's performance on the validation data would assuage any concerns about the importance of these design choices when applying the algorithm to new data. I would prefer to see accuracies measured on validation data over details about the kurtosis of the fit residuals or the rotation of the gear.

The reviewer is correct that photobleaching can potentially present issues for parameters that incorporate the amplitude of the particle images, as we mention on page 8. For this reason, we also present an additional option for axial tracking based on the parameter ρ_w , which does not depend on image amplitude or any normalization.

Our validation of the method by tracking the rotating gear does indeed demonstrate that any photobleaching does not produce significant errors in axial tracking, as shown by the residuals of the rigid transform in the axial direction (Supplementary Figure S14c). These residuals do not increase over time, and the motion of the gear does not exhibit any effects over the course of the measurement series that would be consistent with growing errors in axial localization, such as from photobleaching, as can be seen in Figure 8. We have revised the manuscript to more clearly indicate that any photobleaching results in insignificant errors.

In addition to this main concern, I have several comments of lesser importance:

* The authors should be more clear about the size of the particles used. In the caption to Fig. 1, the authors say that the particles are 1 micrometer. However, they do not say this in the main text, and they do not say what size the emitters are for the aperture arrays or the gear. This is important, as variation in particle sizes may change both fitted amplitude (via changes in the emission intensity) and the fitted Gaussian widths (as the particles are not sub-resolution).

We have moved these and other details to the Methods section of the main text, but also wish to respectfully note that all such information was present in the Supplementary Methods. We have also revised the main text to emphasize the importance of using the same emitters for both calibration and experiment, as we have done in this work.

* The discussion on pg 10 regarding the fitting of the calibration functions is confusing. The authors first state that they fit $z(\rho_a)$, then $x'(z)$, then $z(x', y', \rho_a)$. Are the authors fitting an axial displacement which depends on both the in-plane position and the normalized correlation? Or are the authors just fitting the axial displacement to the normalized correlation?

The reviewer's first description of this aspect of the method is correct. We summarize the complete localization procedure and the role of each model in Table S2 and Figure S2. We model both the

axial dependence of image shape as well as the lateral dependence of this axial dependence. $z(\rho_a)$ models the axial dependence of image shape only at the (x,y) locations of the calibration particles. Because $z(\rho_a)$ changes with lateral position (x,y) , we require an additional model to capture this lateral dependence so that accurate axial tracking is possible at any (x,y) position in the imaging field. This is further complicated by the fact that lateral position erroneously appears to change with axial position, and so we perform an analogous calibration for this apparent lateral motion.

* On pg 7, the authors state that their bivariate Gaussian "approximates the image loci as ellipses with axes at an angle of $\pi/4$ radians with respect to the x and y axes." This is only true if w_x is restricted to be equal to w_y . The functional form used in the equation on pg 5 provides a full description of the covariance matrix and thus allows the axis to be oriented at an arbitrary angle to the x - or y - axes.

We have clarified this point to say that for our particular system, the elliptical axes are at an angle of $\pi/4$ relative to the axes of the imaging sensor. This is why the parameter ρ is particularly useful in this work. We agree with the reviewer that the bivariate Gaussian model is capable of capturing asymmetry along any arbitrary angle and is thus generally applicable to implementation of our tracking method on any system.

* The "astigmatic defocus parameter" $\rho_w = \rho * (|w_x| + |w_y|) / 2$ is not geometrically invariant. For a system with astigmatism extrinsically added in, it makes sense to use a defocus parameter specific to a coordinate frame, since the astigmatism axis is known. For this system without a preferred coordinate system, it might be better to use a geometrically invariant measure. This is a detail, and not especially important, but it is choices like this that make me want a clearer demonstration on a validation set outside of the calibration data.

Our choice of parameters is largely empirical, and we have characterized those that provide useful information for axial localization with our microscope system. While different combinations of gaussian parameters may be optimal for different microscope systems, the concepts our method presents remain directly applicable.

* The authors state that they use a 16-order polynomial to model the z -dependence of the Gaussian parameters, and a 5th-order to choose z_f . A sentence explaining why these orders were chosen might be helpful.

We have revised the manuscript to state that the orders we choose for these polynomial models are necessary to produce quasi-normal residuals. For determination of best focus, a fifth-order polynomial reliably models the curvature of the data such as in Figure 2d, while models of lower order do not. Conceptually, any empirical model that accurately fits the trends in these curves should provide similar performance to the polynomial model we use.

REVIEWER COMMENTS

Reviewer #1 (Remarks to the Author):

The resubmitted version clarifies most issues I had with the first version of the paper. The novelty of the paper lies only in the accurate localization of the marker, which is an absolute position measurement. The location of the marker relatively to the device must be known. However, the topography data obtained for example in a white-light interferometer allows also extracting displacement angles and positions and it is calibrated by the magnified pixel size and the z-axis displacement measurement. Therefore, this is not the first metrology tool for 6 degrees of freedom measurement of MEMS. The authors still rather give the impression of this claim but it is not written directly. Thus, I think it can be published but I like to ask the authors to remove "in Six Degrees of Freedom" from the title which I found confusing.

Reviewer #2 (Remarks to the Author):

The authors have sufficiently addressed my concerns.

REVIEWER COMMENTS

We thank both reviewers for their time and consideration.

Reviewer #1 (Remarks to the Author):

The resubmitted version clarifies most issues I had with the first version of the paper. The novelty of the paper lies only in the accurate localization of the marker, which is an absolute position measurement. The location of the marker relatively to the device must be known. However, the topography data obtained for example in a white-light interferometer allows also extracting displacement angles and positions and it is calibrated by the magnified pixel size and the z-axis displacement measurement. Therefore, this is not the first metrology tool for 6 degrees of freedom measurement of MEMS. The authors still rather give the impression of this claim but it is not written directly. Thus, I think it can be published but I like to ask the authors to remove "in Six Degrees of Freedom" from the title which I found confusing.

We have removed "in Six Degrees of Freedom" from the title.

Reviewer #2 (Remarks to the Author):

The authors have sufficiently addressed my concerns.

Thank you.